# Effects of Long-Term Exposure to Cadmium on Development, Reproduction and Antioxidant Enzymes of *Aleuroglyphus ovatus* (Acari: Acaridae)

**DOI:** 10.3390/insects13100895

**Published:** 2022-09-30

**Authors:** Yu Zhang, Wenhui Xiong, Shan Yang, Hui Ai, Zhiwen Zou, Bin Xia

**Affiliations:** 1School of Life Science, Nanchang University, Nanchang 330031, China; 2Institute of Life Science, Nanchang University, Nanchang 330031, China

**Keywords:** *Aleuroglyphus ovatus*, cadmium, development, reproduction, antioxidant enzymes

## Abstract

**Simple Summary:**

Cadmium is one of the major metal pollutants in grain, threatening food safety and human health. There is an urgent need to find an organism that is prevalent in grain and which can be used as a biological model for determining and assessing the effects of long-term heavy metal contamination in offspring. The findings in this study showed that long-term cadmium exposure could adversely affect the development, reproduction and physiology of *Aleuroglyphus ovatus* (Acari: Acaridae). *A. ovatus* was sensitive to cadmium, which could be a valuable tool for studying the toxicity of long-term heavy metal exposure on offspring. Simultaneously, there is little report about this study, providing a basis for future evaluation of cadmium pressure on genetic evolution.

**Abstract:**

Grain contaminated by cadmium (Cd) has become a serious food security problem, and it is necessary to determine and evaluate the toxic effect and defense mechanism of long-term heavy metal pollution in grain. In order to evaluate the effects of long-term heavy metal Cd stress on the stored grain pests, *Aleuro**glyphus ovatus* were fed with an artificial diet supplemented with different concentrations of Cd (0, 5, 10, 20 mg/kg). The development, fecundity and detoxification enzymes of *A. ovatus* were analyzed and observed. In this study, the immature duration of *A. ovatus* was significantly prolonged under long-term Cd stress. Moreover, the survival duration of female adults was significantly shortened. The total number of eggs laid and the daily number laid per female adult decreased significantly. There were significant differences in protein content at protonymph and tritonymph stages when the concentration of Cd exceeded 10 mg/kg. The protein content of female adults was higher than that of male adults. The activity of detoxification enzymes showed differences in different conditions, such as development stage, Cd concentration and gender. These findings confirmed that *A. ovatus* were sensitive to Cd, and their offspring were severely affected under long-term Cd stress. Therefore, *A. ovatus* is a good model for evaluating the toxicity of long-term heavy metal Cd stress. The study provides the basis and enriches the research content of heavy metal pollution on mites, contributing to the harmonious and healthy development between the environment and human beings.

## 1. Introduction

With the continuous development of industrialization and urbanization, heavy metal pollution has become a global environmental problem. Heavy metals have the characteristics of concealment, permanence and irreversibility, which threaten biodiversity and accelerate the deterioration of ecology. Cadmium (Cd) is a heavy metal element in nature, which can be detected in a wide range of foods. The Joint FAO/WHO Expert Committee on Food Additives (JECFA) reviewed foods distributed in 28 countries and discovered that the mean cadmium concentrations ranged from 0.1 to 4.8 mg/kg [1]. However, cadmium can also be accumulated at high levels of concentration and is toxic throughout the food chain. Cd is one of the major metal pollutants in the world, affecting not only the yield and quality of the grain but also human health through the respiratory tract, alimentary canal, skin and food chain [2,3,4]. When it accumulates to a certain extent in the body, Cd directly causes severe injuries to the human body, such as hepatic toxicity, renal dysfunction and genotoxicity [5].

The toxic effects of heavy metals on insects and invertebrates have drawn the attention of researchers. Long-term exposure to Cd could alter the development of black soldier fly larvae and *Aedes albopictus* [6,7]. Both the development and reproduction of *Oncopeltus fasciatus* are seriously affected at sublethal Cd concentrations [8]. Offspring characteristics were also focused on for further evaluating the effects of exposure to environmental toxic substances. The second-generation effects of metal pollutant exposure on reproduction, longevity and insecticide tolerance have been reported [9]. The most frequently observed effects were epigenetic modifications as well as changes in fertility and hatchability, which were summarized in a review on the responses of insects to different toxicants [10]. Some metal ions accumulated in insects would change the activities of detoxifying enzymes [11]. Many studies show that Cd could induce oxidative stress and reduce the activity of antioxidative defense system enzymes. Among various mechanisms of antioxidant defense enzymes, superoxide dismutase (SOD), peroxidase (POD), glutathione transferase (GST) and catalase (CAT) play an essential role in defending against Cd toxicity [12,13,14].

*Aleuro**glyphus ovatus* (Acari: Acaridae) is an economic pest of stored food products worldwide [15]. They are commonly found in stored grains such as wheat, flour and corn. As an essential component of biodiversity, mites are associated with national food security and the structure of the ecological environment. In recent years, it has been reported that the amount of heavy metals exceeds the standard in stored grain and diet [16,17,18]. The transfer and cumulative effect of heavy metals in an ecosystem is a hot issue in studying ecological environment damage caused by heavy metal pollution. Meanwhile, *A. ovatus* play an essential role in the accumulation and transportation of heavy metals, attributing to the characteristics of small individuals, fast reproduction rate and short generation cycle.

There are few studies about the influence of cadmium on *A. ovatus*. We are interested in the ability of mites that originated from grain storage to cope with high concentrations of cadmium under long-term exposure. This study also considered whether *A. ovatus* could be used as a model for evaluating the toxicity in stored grain contaminated by heavy metals. It is of great significance to study the effects of heavy metal Cd on the survival and reproduction of offspring, which aims at protecting the diversity of species, stability of the ecosystem and maintenance of human health.

## 2. Materials and Methods

### 2.1. Mites Culture

*Aleuro**glyphus ovatus* (Acari: Acaridae) was initially obtained from a provender mill in Wangcheng town, Xinjian county of Jiangxi province, China. Mites were cultured in artificial climate chambers (MGC-800HP-2, Shanghai Yiheng Scientific Instrument Corporation, Shanghai, China) under constant conditions (26 ± 2 °C, 80 ± 5% RH, 0L:24D). *A. ovatus* was, respectively, fed on artificial diets of different concentrations (0,5,10,20 mg/kg) of Cd for six months as research materials. The artificial diets were composed of wheat flour (COFCO Corporation, Beijing, China) and yeast extract (Angel Yeast Corporation, Yichang, China) in a 4:1 mass ratio and mixed with a certain amount of distilled water. The cadmium content of flour and yeast extract were lower than 0.1 mg/kg (GB 2762-2022). Furthermore, 1 mg/mL of cadmium chloride (CdCl_2_) was added to the artificial diet according to the experimental design for treatment groups (5, 10, 20 mg/kg). Analytical-grade cadmium chloride was purchased from Shanghai Chemical Industry Park (Shanghai, China). All the mixtures were dried naturally and then crushed with a feed mincer (GR-258b, Rongshida Appliance Corporation, Hefei, China).

### 2.2. Developmental Durations

Eggs produced within 12 h were transferred to the plexiglass plates with holes; they were collected from control (0 mg/kg) and treatment (5, 10, 20 mg/kg) groups with an extra fine brush. The dimension of each plexiglass is 300 mm × 100 mm × 8 mm (length–width–height). One egg was placed in each hole (8 mm × 3 mm diameter-depth), and cover glasses were sealed on the holes with the glycerine–water mixture. One hundred eggs were collected for each sample, and the control and treatment groups had three samples, respectively. All the plexiglass plates were placed in the artificial incubator with set conditions. Meanwhile, a certain amount of artificial diets ensured adequate food for mites in control and treatment groups every 3 days. The duration of development at different stages was recorded at 12 h intervals with a stereoscope (Olympus SZ61, magnification 45×, Tokyo, Japan). 

### 2.3. Reproductive Evaluation

The *A. ovatus* that newly entered the adult stage were selected from control and treatment groups and transferred to the corresponding plexiglass plates with holes. One *A. ovatus* female adult was placed in a hole, then two male adult mites (0 mg/kg) were added to each hole for matching. New male adults must be added in time if the male adult mites die. The duration of total oviposition was recorded at 12 h intervals until all female adults died. By taking thirty female adults as a sample, control and treatment groups had three samples, respectively. The eggs produced were continuously recorded and removed by a stereoscope (Olympus SZ61, magnification 45×) every 24 h.

### 2.4. Determination of Protein Content and Enzyme Activity

The samples of *A. ovatus* required from different Cd treatment groups were 1000 eggs (within 24 h), 800 larvas (the first instar), 600 protonymphs (the first instar), 400 tritonymphs (the first instar), 200 female adults (2 days) and 200 male adults (2 days), respectively. Samples were collected separately in 1.5 mL centrifuge tubes and stored in a refrigerator at −80 ℃ for later use. The protein content of each sample was determined by Protein Assay Kit (Sangon, C503031). All the samples were weighed and homogenized. The test products were obtained by centrifugation (Eppendorf 5417R, 9391 g, 4 °C, 5 min) before determining protein content. The activity of SOD, POD, CAT and GST were separately determined using the Superoxide Dismutase Activity Assay Kit (Micromethod, D799594), Peroxidase Activity Assay Kit (Micromethod, D799592), Catalase Activity Assay Kit (Micromethod, D799598) and Glutathione S-transferase Activity Assay Kit (Micromethod, D799612) following the instructions. All the assay kits were purchased from Sangon Biotech (Shanghai, China). The experiments were performed in three triplicates. 

### 2.5. Statistical Analysis

The effects of Cd stress on the average duration at different developmental stages, oviposition duration, fecundity, protein content and detoxification enzymes were determined using the one-way analysis of variance (ANOVA) and further analyzed by Tukey’s test. Egg hatchability and mortality rate were both analyzed using the Chi-squared test. All the data analyses were performed using IBM SPSS Statistics software (version 20, International Business Machines Corporation, New York, NY, USA), and *p* values below 0.05 indicated a significant difference.

## 3. Results

### 3.1. Effects of Cadmium on the Development

The development of *A. ovatus* under long-term Cd stress at different concentrations is shown in Figure 1. Compared with the control group (0 mg/kg), the egg hatchability of *A. ovatus* was significantly decreased under different Cd concentrations (Figure 1a). With the increase in Cd concentration, the mortality rate of all developmental stages increased, and the highest death rate occurred at 20 mg/kg (Figure 1b). Under long-term Cd stress at different concentrations, the immature duration of *A. ovatus* was significantly prolonged. The maximum value was 27.65 days at 20 mg/kg, 1.55 times higher than the control group (Figure 1c). The major immature stage of *A. ovatus* includes egg, larva, protonymph and tritonymph. The larva, protonymph and tritonymph of *A. ovatus* occurred significantly later in Cd treatment groups than in the control group (Figure 1d). It should be noted that the eggs of *A. ovatus* hatched earlier at 5 mg/kg compared with the control group (0 mg/kg) and other treatment groups.

### 3.2. Reproductive Evaluation

Long-term exposure to Cd affected the oviposition duration of *A. ovatus* (Table 1). The pre-oviposition duration of *A. ovatus* was significantly prolonged with the increase in Cd concentration. However, the oviposition duration increased significantly under the long-term Cd stress of 5 and 10 mg/kg. Although there was no difference between the 20 mg/kg Cd treatment and the control group (0 mg/kg), the survival duration was significantly shortened at 5 and 20 mg/kg after the oviposition stage. Female adult mites’ lifespan was significantly longer than the control group at 10 mg/kg. Both the total number of eggs laid and the daily number laid per female adult decreased significantly with the increase in Cd concentration (Figure 2). Compared with the control group, the total number of eggs laid by each female decreased from 152.76 to 47.82 at 20 mg/kg cadmium stress. Meanwhile, the number of eggs laid per female decreased from 4.93 eggs/d to 1.45 eggs/d.

### 3.3. Effects of Cadmium on Protein Content

The protein content of *A. ovatus* at different developmental stages under different Cd concentrations is shown in Figure 3. Compared with the control group (0 mg/kg), there was no significant difference in protein content at the stages of egg and larva with the increase in Cd stress concentration (Figure 3a). However, when the concentration of Cd increased to 10 mg/kg, there were significant differences at the protonymph and tritonymph stages compared with the control group. Especially when the Cd stress concentration reached 20 mg/kg, the protein content of *A. ovatus* was the maximum at the stage of protonymph and tritonympht; they were 1.22 times and 1.45 times that of the control group, respectively. Under the same Cd concentration of stress, the protein content of the egg, juvenile, protonymph and tritonymph showed an apparent increase. However, there was no significant difference between larva and protonymph. Compared with the control group, the protein content of female or male adults showed no significant difference with the increase in Cd concentration (Figure 3b). However, the protein content of both female and male adults increased significantly when Cd concentration reached 20 mg/kg compared with the control group. The protein content of female adults was higher than that of male adults under the same Cd concentration of stress.

### 3.4. Effects of Cadmium on Activities of Detoxification Enzyme

The activity variance of *A. ovatus* detoxification enzymes at different immature development stages was performed under different concentrations of Cd stress (Figure 4). The analysis exhibited that the SOD activity at different immature development stages was higher than in the control group when the Cd concentration reached 5 mg/kg (Figure 4a). When Cd concentration reached 10 mg/kg, the SOD activity of nymphs and adult mites decreased significantly with the increase in Cd concentration (Figure 4a and Figure 5a). Exposure of *A. ovatus* to Cd decreased POD and CAT activity at all immature development stages compared to the control group (Figure 4b,c). GST activity showed no significant difference at the same immature development stage among 5 and 10 mg/kg treatment groups, in addition to the egg stage (Figure 4d). Moreover, 20 mg/kg GST activity was the lowest in all immature stages compared with other treatment concentrations.

The SOD activity of male adults in all treatment groups was significantly higher than that of female adults (Figure 5a). Meanwhile, male adults’ SOD, POD and CAT activity were higher than female adults when Cd concentrations reached 10 and 20 mg/kg (Figure 5b,c). With the increase in Cd concentration, the GST activity of both female and male adults decreased significantly (Figure 5d). Nevertheless, there was no significant difference between female and male adults under the same Cd concentration stress. Detoxification enzyme activity of different development stages and genders had noticeable differences under different concentrations of Cd stress.

## 4. Discussion

The transfer and accumulation of Cd in different organisms through the food chain may play a significant role in negatively regulating the development and metabolism of insects. Merrington et al. [19] observed the uptake and accumulation of Cd in *Aphis fabae* and discovered a potential transfer route from wheat to aphids. *Tetrix tenuicornis* were collected from a polluted area, and the Cd level was many times higher than that collected from unpolluted areas [20]. There is literature on the effects of insects exposed to Cd, yet the research taken up on mites remains scarce. Therefore, our study is possibly the first to investigate the effect of long-term Cd exposure on the development, reproduction and physiology by feeding it to *A. ovatus*.

By comparing the growth of *A. ovatus* fed with untreated artificial diets, it was observed that Cd slowed down the growth and prolonged the development duration. Our study indicated that the development duration of *A. ovatus* was prolonged by 9 days at 20 mg/kg under long-term stress. The result is also consistent with Cd fed to other insects. It showed that Cd could slow down the growth and development of *Ostrinia nubilalis*, which accumulated from diets at 41.7 mg/kg after long-term exposure [12]. Perhaps one of the possible reasons for the delay is the energy shortage caused by metals. The development and growth need energy obtained from the metabolism of carbohydrates, lipids and proteins [6,21]. It is worth mentioning that our study also confirmed similar results. The protein content result showed an apparent increase in *A. ovatus* at 20 mg/kg. The organism needs to store more energy for detoxification to prevent metal damage [8,22]. Furthermore, the energy metabolism caused by metals should be paid more attention to in the future.

Feeding insects with diets contaminated by heavy metals can negatively affect reproduction [23,24]. In our study, Cd could prolong the pre-oviposition and oviposition duration of *A. ovatus* but seriously reduced the fecundity of Cd-treated females under long-term stress. Fecundity represents the reproductive function and ability of animals to produce offspring, defined as the total number of offspring laid per female during the whole oviposition period [25]. The fecundity of *Aedes albopictus* adults could be significantly reduced with the increased concentration of Cd [7]. The oviposition rate of *A. ovatus* females was also significantly affected by Cd exposure in our study, which was almost three times higher in the control group than treatment group (20 mg/kg). Moreover, the duration of *Spodoptera exigua* female lifespan in the Cd strain is shorter than individuals from the control. The results are consistent with the findings on *Spodoptera exigua* in a previous study [26]. Our results showed that the reproduction of *A. ovatus* was sensitive to cadmium; one possible reason was the development of slower with longer growth cycles. Cd could also interfere with calcium channels in the central nervous system (CNS) and disturb hormones in regulating ovarian development [24]. We assumed that long-term Cd exposure might affect ovarian function, damage oocyte structure and change the expression of genes relating to development or reproduction in offspring. These speculated reasons may cause reproductive toxicity of the ovary and maternal toxicity of offspring development.

Many studies indicated that heavy metals would promote the formation of ROS (reactive oxygen species) [27,28,29]. The balance between production and elimination of ROS is kept under normal circumstances, in which extra heavy metals would strike due to the variation in antioxidant enzyme activities [30]. Cd influenced antioxidant enzyme activity under long-term stress in our study, which supports the results in previous research that showed changes in the activity of SOD, POD, CAT and GST. In order to adapt to long-term heavy metal pollution, insects may evolve unique physiological and biochemical detoxification functions to reduce the harm of excessive heavy metals and maintain the homeostasis of the organism’s internal environment. It was reported that SOD plays an important role in working on the front line of the defense system against oxidative stress [31,32]. The SOD activity was observed in housefly larvae and decreased after 24 h exposure to 20 mM Cd [5]. According to the previous study, Cd treatment could change the structure of SOD protein and decrease SOD activity. When Cd concentration reached 10 mg/kg, our study showed that the SOD activity of nymphs and adult mites decreased significantly. The possible reason is that Cd pressure could reduce the content of Zn^2+^, which play a critical role in maintaining the conformation of SOD protein [33]. Ahmad et al. [34] suggested that POD and CAT could function by removing H_2_O_2_. The activity of both POD and CAT in *A. ovatus* was negatively affected by Cd during the whole development stage. Our results showed Cd would decrease the activity of enzymes and implied detoxification mechanism may have lost its efficacy. Remarkably, the activity of SOD, POD and CAT in male adults was higher than that of female adults when Cd concentrations reached 10 and 20 mg/kg. GSTs play an essential role in detoxifying xenobiotics by catalyzing nucleophilic attack and reducing oxidative damage [35]. The present study showed that the GST activity at some immature stages was statistically significant, which may have contributed to life-stage-specific physiological characteristics. Similar results were found in *Oxya chinensis* and *Poecilus cupreus* [35,36]. The increased Cd concentration significantly inhibited the GST activity of *A. ovatus* in female and male adults. However, the significant inhibition only occurred in the GST activity of *Drosophila melanogaster* females [23]. These different results were probably due to the variety of species, accumulation of metal and different genders.

Based on these findings, we found that long-term Cd exposure had adverse toxicity effects on the development, fecundity and detoxification enzymes of *A. ovatus*. Meanwhile, the mechanism of our findings needs further research in the future. This study provides a scientific basis for further research on the toxic effect and mechanism of long-term Cd exposure on *A. ovatus*. At the same time, it is confirmed that the ecological risk assessment of Cd pollution should pay attention to its influence on offspring.

## Figures and Tables

**Figure 1 insects-13-00895-f001:**
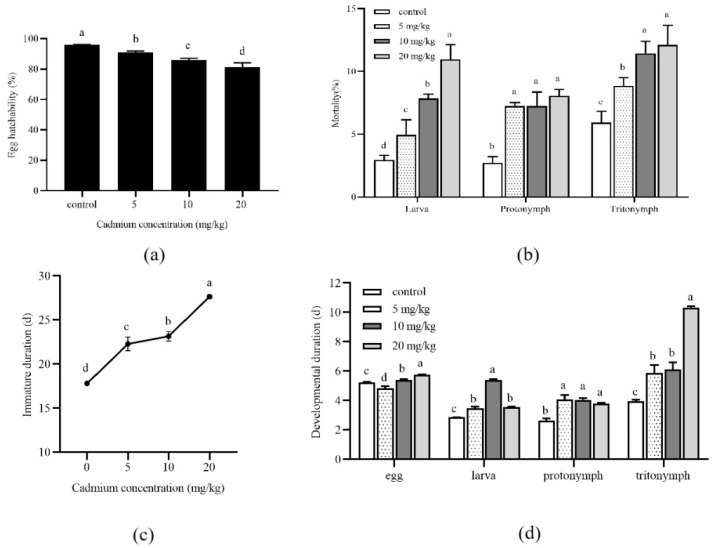
Effects of long-term cadmium exposure on the development of Aleuroglyphus ovatus. (**a**) Egg hatchability; (**b**) Mortality of different development stages; (**c**) Total immature development duration; (**d**) Development duration of major immature stages. Results are expressed as mean ± SD. Different lowercase letters indicate significant differences between treatments at different Cd concentrations (*p* < 0.05).

**Figure 2 insects-13-00895-f002:**
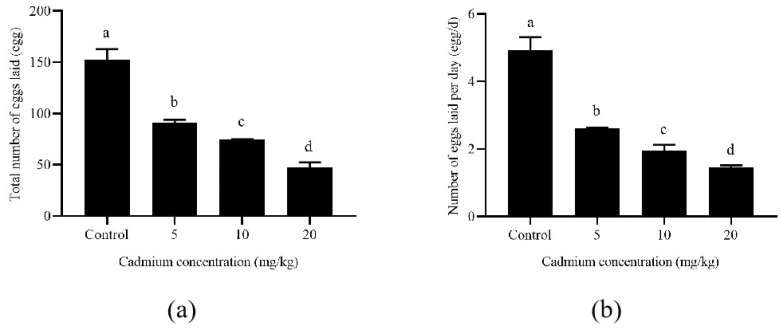
The number of eggs laid per *Aleuro**glyphus ovatus* female adult under different Cd concentration stress. (**a**) The total number of eggs; (**b**) The Number of eggs laid per day. Results are expressed as mean ± SD. Different lowercase letters indicate significant differences among control and treatment groups at different cadmium concentrations (*p* < 0.05).

**Figure 3 insects-13-00895-f003:**
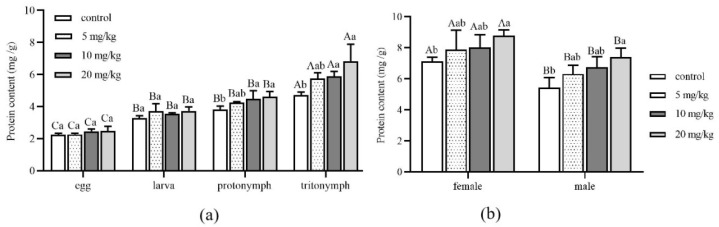
The protein content of *Aleuroglyphus ovatus* under different concentrations of Cd stress. (**a**) Different immature development stages; (**b**) Female and male adults. Results are expressed as mean ± SD. Different uppercase letters indicate the significant differences among major immature development stages at the same Cd concentration (*p* < 0.05). Different lowercase letters indicate significant differences among control and treatment groups at the same development stage (*p* < 0.05).

**Figure 4 insects-13-00895-f004:**
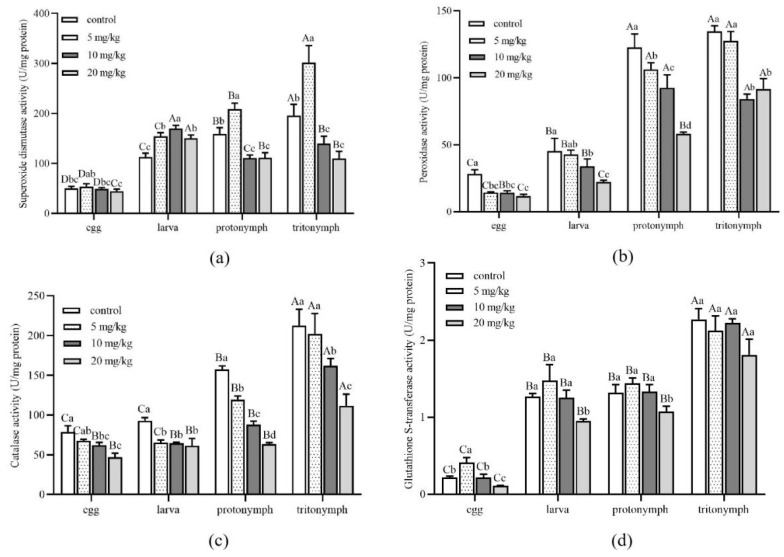
Detoxification enzyme activity of *Aleuroglyphus ovatus* under different concentrations of Cd stress at different immature development stages. (**a**) Activity of superoxide dismutase (SOD); (**b**) Activity of peroxidase (POD); (**c**) Activity of catalase (CAT); (**d**) Activity of glutathione S-transferase (GST). Results are expressed as mean ± SD. Different uppercase letters indicate significant differences among major immature development stages at the same Cd concentration (*p* < 0.05). Different lowercase letters indicate significant differences among control and treatment groups at the same development stage (*p* < 0.05).

**Figure 5 insects-13-00895-f005:**
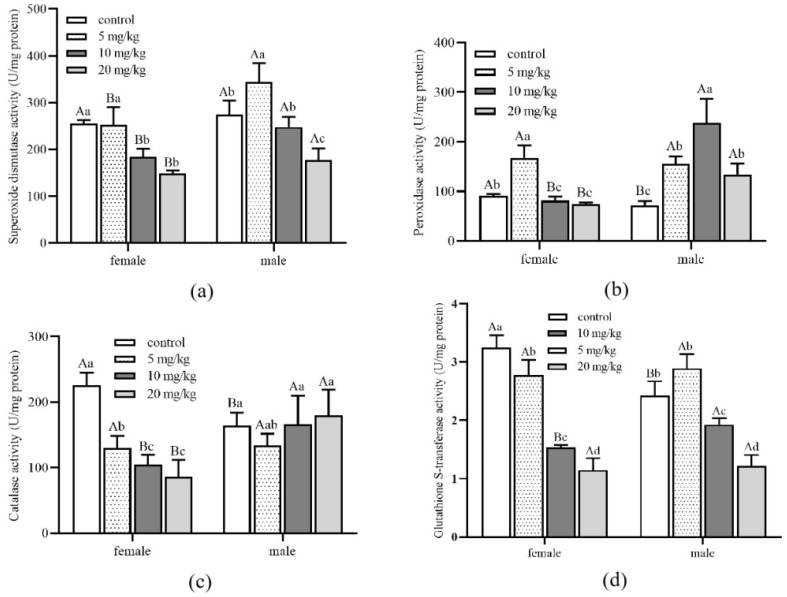
Detoxification enzyme activity of *Aleuroglyphus ovatus* under different concentrations of Cd stress in female and male adults. (**a**) Activity of superoxide dismutase (SOD); (**b**) Activity of peroxidase (POD); (**c**) Activity of catalase (CAT); (**d**) Activity of glutathione S-transferase (GST). Results are expressed as mean ± SD. Different uppercase letters indicate significant differences between female and male adults at the same Cd concentration (*p* < 0.05). Different lowercase letters indicate significant differences among control and treatment groups in female and male adults (*p* < 0.05).

**Table 1 insects-13-00895-t001:** Effects of different cadmium concentrations on the oviposition duration of *Aleuroglyphus ovatus*. Different lowercase letters indicate significant differences among control and treatment groups at different cadmium concentrations (*p* < 0.05).

Duration (d) (Mean ± SD)	Cadmium Concentration (mg/kg)			
	0	5	10	20
Pre-oviposition period	2.45 ± 0.13 c	4.52 ± 0.21 b	5.15 ± 0.10 a	5.62 ± 0.09 a
Oviposition period	31.34 ± 1.94 bc	34.62 ± 0.44 ac	37.64 ± 1.48 a	32.36 ± 0.79 bc
After the oviposition period	10.90 ± 1.19 a	6.95 ± 0.48 cd	9.09 ± 0.44a b	8.55 ± 0.05 bc
Lifespan of female adult	44.69 ± 2.71 bc	46.08 ± 0.55 ab	50.76 ± 1.37 a	46.01 ± 1.34 ab

## Data Availability

The data presented in this study are available on request from the authors.

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
