# Peer review of "Effects of Long-Term Exposure to Cadmium on Development, Reproduction and Antioxidant Enzymes of Aleuroglyphus ovatus (Acari: Acaridae)"

_insects, 2022, doi:10.3390/insects13100895_

Round 1
Reviewer 1 Report (Previous Reviewer 1)
The review manuscript by Zhang et al “Effects of long-term exposure to cadmium on development, reproduction and antioxidant enzymes of Aleuroglyphus ovatus (Acari: Acaridae)” presents results that suggest cadmium delays development and reduces reproduction in this flour mite. I recommend publication pending minor review.
Line 90: Which type of flour?
Line 94: ‘Mother liquid’ Appropriate wording needed
Line 99, Section 2.2: The experiment set up is still not clear. How were the female adults selected for egg laying? What are the dimensions of the Plexiglass? How many holes and size per Plexiglass? Were eggs placed in all holes per Plexiglass? How many eggs per treatment? What was used to transfer the eggs? Did the holes contain artificial diet and how much? More details on how monitoring and data collection were carried out from egg to adult stage. Assuming the mite was inside the flour how was it located for data recording?
Line 109, Section 2.3: The experimental details are not clear. How many mites were selected per treatment? Were these held in Plexiglass? Was the experiment repeated and how many times?
Line 117, Section 2.4: A brief explanation how proteins were extracted and determined, and how enzyme activity was measured should be added apart from just citing the kits.
Lines 119 -120: Why first instar in brackets for three different stages?
Line 289: ‘Critical member who works’ – Appropriate wording needed
Author Response
Dear Reviewer,
Thanks very much for taking your time to review our manuscript. We really appreciate all your comments and suggestions! We have carefully considered your suggestions and tried our best to revise our manuscript according to the comments.
Please see the attachment.

Reviewer 2 Report (Previous Reviewer 2)
The authors revised the manuscript extensively and now the story reveals itself in the new version. Cd contamination is no doubt one of the biggest problems in massive grain storage. The manuscript utilized a flour mite to have investigated the effect of long-term exposure of Cd to the development and reproduction, and a conclusion was claimed following the investigation: this mite was sensitive and was expected to serve as an indication model for evaluating Cd toxicity in stored grain. The results are found sound, and the conclusion stands with the support of adquate data. Overall, I find this version acceptable.
Author Response
Dear Reviewer,
Thanks very much for taking your time to review our manuscript. We really appreciate all your comments and suggestions! We have carefully considered your suggestions and tried our best to revise our manuscript according to the comments.
Please see the attachment.

Reviewer 3 Report (New Reviewer)
The manuscript reported the effects of Cd long-term exposure on Aleuroglyphus ovatus, focusing on its consequences on development, reproduction and detoxification enzymes of the Acaridae. I found this research very interesting, a lot of work was done and explain clearly. However, I suggest to re-arrange the discussion section since it is a little chaotic and the sentences seems disconnected one to each other. Moreover, a conclusion that underlying distinctly the importance of your findings is needed.
L. 22: a “with” is lacking after “fed”
INTRODUCTION
L.51-52: More attention than what? Rewrite the sentence because it is not clear
L.53: examples of the insects mentioned would be useful
L.55-56: a verb seems to lack
L.71:74. Rewrite explaining better why it plays an essential role
MATERIALS AND METHODS
L.85: What was the diet of the mites in the breeding culture?
L.88: Why 20 mg/kg was chosen as maximum concentration?
RESULTS
L.153: here, and in the rest of the manuscript, you write “error bars” but reported that they are standard deviations. Is it right?
L.160: this kind of comments should go in the Discussion section
L.164:167: specify to which treatment those values belong
Discussion should be re-write, many verbs or conjunctions are lacking. Moreover, many examples are inappropriate since they refer to different species AND metals. There are a few careless mistakes like in line 263, where you refer to a species that is not the subject of your manuscript. Finally, from L.278 the manuscript results confused. Too many examples of other species are added without a solution of continuity. It is hard to understand when the authors are talking about their own work or not, and beginning with a sentence that contrasts your findings should clearly underly this contrast, while here this concept does not emerge.
Author Response
Dear Reviewer,
Thanks very much for taking your time to review our manuscript. We really appreciate all your comments and suggestions! We have carefully considered your suggestions and tried our best to revise our manuscript according to the comments.
Please see the attachment.

This manuscript is a resubmission of an earlier submission. The following is a list of the peer review reports and author responses from that submission.
Round 1
Reviewer 1 Report
The review manuscript by Zhang et al “Effects of long-term exposure to cadmium on development, reproduction and antioxidant enzymes of Aleuroglyphus ovatus (Acari: Acaridae)” presents results that suggest cadmium delays development and reduces reproduction in this flour mite.
This manuscript requires extensive editing of English language and style. There numerous grammar errors such as misspelled words, missing words in sentences and poor word choice.
The concept of this study is not clear. Why study the effect of cadmium contamination on the mite? The mite cannot in any way be used to determine cadmium contamination without laboratory tests to confirm presence and concentration of cadmium. The authors need to clarify the value of this study. In its current form it is too basic to qualify for publication in Insects.
The materials and methods section must be improved. So many details are lacking making it difficult to understand how experiments were done.
Line 95: Describe what binocular solid lens are and magnification used
Line 106/107: Why repeated first instar in brackets for different stages of nymphs?
Figure 2: Based on the proportions of the columns select and present either total eggs or eggs per day but not both otherwise it is confusing.
Results: It is difficult to interpret the results because it is not clear how these were obtained in materials and methods.
Discussion: It has lots of sections that are more of literature review than discussing the results. It is not clear how this mite can be used as indicator for heavy metal contamination. Flour may contain other components that can affect flour mite reproduction and survival. This study mentions artificial diet with no details of what it comprised. It would be appropriate to include experiments using flour samples of known cadmium/heavy metal concentrations.
Author Response
Dear Reviewer,
Thanks very much for taking your time to review our manuscript (Manuscript ID: insects-1830806). We really appreciate all your comments and suggestions! We have carefully considered your suggestion and tried our best to revise our manuscript according to the comments.
All changes have been highlighted in new version of manuscript and the following are the responses.
Response to the comments of Reviewer #1
Comment No.1: This manuscript requires extensive editing of English language and style. There numerous grammar errors such as misspelled words, missing words in sentences and poor word choice.
Response: The manuscript has been checked by a native speaker.
Comment No.2: The concept of this study is not clear. Why study the effect of cadmium contamination on the mite? The mite cannot in any way be used to determine cadmium contamination without laboratory tests to confirm presence and concentration of cadmium. The authors need to clarify the value of this study. In its current form it is too basic to qualify for publication in Insects.
Response: I apologize for making the concept of the article unclear. Based on the reviews of the experts, the value of this article is rewritten. Cadmium is one of the major metal pollutants in grain which threatens food safety and human health. There is an urgent need to find an organism that is prevalent in grain and which can be used as a biological model for determination and assessment the effects of long-term heavy metal contamination on offspring. Therefore, the study provides the basis and enriches the research content of heavy metal pollution in stored grain, contributing to the harmonious and healthy development between the environment and human beings. (New version of manuscript, in line 12-19 and line 33-35)
Comment No.3: The materials and methods section must be improved. So many details are lacking and making it difficult to understand how experiments were done.
Response: It is really true as reviewer mentioned that due to incompleteness of our materials and methods section. So we have improved and supplemented this content in detail. ( New version of manuscript, in line 90-103 and line 122-125)
Comment No.4: Line 95: Describe what binocular solid lens are and magnification used.
Response: I'm sorry for the misrepresentation of the instrument. It is a stereoscope (Olympus SZ61, magnification 45x).
Comment No.5: Line 106/107: Why repeated first instar in brackets for different stages of nymphs?
Response: Experiment require a large number of samples, selecting first instar of nymphs at different stages is the most accurate and easy way to obtain samples. Aleuroglyphus ovatus is a small mite, whose development stages includes egg, larva, the first resting stage, protonymph, the second resting stage, tritonymph, and the third resting stage. The resting period is relatively short, but the mites are static and it is convenient to observe and pick, so they can be directly picked out as samples when they develop into the first day of nymphs for different stages.
Comment No.6: Figure 2: Based on the proportions of the columns select and present either total eggs or eggs per day but not both otherwise it is confusing.
Response: Thank you for your suggestion. We have redrawed the data diagram.(New version of manuscript, Figure 2)
Comment No.7: Results: It is difficult to interpret the results because it is not clear how these were obtained in materials and methods.
Response: It is a pity that both the organization and use of language in this manuscript are not accurate enough to express the materials and methods, which make you unclear at the results. We have improved the materials and methods in details according to your suggestions mentioned above.
Comment No.8: Discussion: It has lots of sections that are more of literature review than discussing the results. It is not clear how this mite can be used as indicator for heavy metal contamination. Flour may contain other components that can affect flour mite reproduction and survival. This study mentions artificial diet with no details of what it comprised.
Response: The production of artificial feed has been supplemented in detail. (New version of manuscript, in line 90-98)
The artificial diet without cadmium (0 mg/kg) was used as control group, and the components of artificial diet are in accordance with the national standards. So other components affecting mite reproduction or survival could not be considered in this study.
Thanks again to the hard work of the reviewer!
Reviewer 2 Report
Despite a seeming attempt of investigating the negative impact of Cd on the development and reproduction of a mite commonly found in stored grains, the configuration, language, grammar and flow of this manuscript do not allow the reviewer to follow, not alone to enjoy a nice story-telling. Besides the above, one of the many but not limited, is that the literature citation, which is exemplified in line 54. This literature reported a transgenerational effect of a single exposure to metal pollution in the malaria vector Anopheles mosquitoes, not an inheritance of developmental toxicity in Drosophila as stated by the authors in the current manuscript.
Overall, the reviewer recommends an extensive polishing throughout the manuscript before any further action can be possibly considered to undertake.
Author Response
Dear Reviewer,
Thanks very much for taking your time to review our manuscript (Manuscript ID: insects-1830806). We really appreciate all your comments and suggestions! We have carefully considered your suggestion and tried our best to revise our manuscript according to the comments.
All changes have been highlighted in new version of manuscript and the following are the responses.
Response to the comments of Reviewer #2
Comment No.1: Despite a seeming attempt of investigating the negative impact of Cd on the development and reproduction of a mite commonly found in stored grains, the configuration, language, grammar and flow of this manuscript do not allow the reviewer to follow, not alone to enjoy a nice story-telling.
Response: We have carefully considered the suggestion of reviewer and tried our best to revise our manuscript according to the comments.
Comment No.2: Besides the above, one of the many but not limited, is that the literature citation, which is exemplified in line 54. This literature reported a transgenerational effect of a single exposure to metal pollution in the malaria vector Anopheles mosquitoes, not an inheritance of developmental toxicity in Drosophila as stated by the authors in the current manuscript.
Response: I apologize for describing uncleanly and citing inaccurately. We have removed it and cited new literature. (New version of manuscript, in line 58-60)
Comment No.3: Overall, the reviewer recommends an extensive polishing throughout the manuscript before any further action can be possibly considered to undertake.
Response: We have carefully considered the suggestion of reviewers and tried our best to revise our manuscript according to the comments. All changes have been highlighted. In addition, the manuscript has been checked by a native speaker.
Thanks again to the hard work of the reviewer!
Reviewer 3 Report
The ms. titled: “Effects of Long-term Exposure to Cadmium on Development,
Reproduction and Antioxidant Enzymes of Aleuroglyphus ovatus (Acari: Acaridae)” provides valuable results on different key consequences of Cd intoxication by arthropods and discuss on the value of using acari as bioindicators of the contamination with this heavy metal. The procedures and protocols applied are well described, developed and presented. I consider this work may turn in to a solid contribution to the field. However, few changes need to be considered before that.
1-This draft needs a rigorous grammar and English-proof revision before its publication. This version does not read easy, and have several writing errors and typos that need to be fixed.
2-I recommend to include and briefly comment on the potential evolutionary consequences, as it is being found both in arthropods and vertebrates (e.g. humans), that the pollution by these heavy metals may trigger epigenetic molecular mechanisms involved in long stress responses. These can have transgenerational consequences, either promoting the adaptation of these organism to the contaminated environment, or participating in teratogenic or pathological responses. In this context I recommend you that when introducing the toxic effects of heavy metals on insects and invertebrates please cite:
Olivares-Castro, G., Cáceres-Jensen, L., Guerrero-Bosagna, C., & Villagra, C. (2021). Insect epigenetic mechanisms facing anthropogenic-derived contamination, an overview. Insects, 12(9), 1–29. https://doi.org/10.3390/insects12090780
And regarding Cd toxic effects on humans:
B. Wang, Y. Li, C. Shao, Y. T. and L. C. (2012). Cadmium and its epigenetic effects. Current Medicinal Chemistry, 19(16), 2611–2620.
I will happily review a corrected version of this work.
Best regards,
Reviewer.
PS: Same comments send to Editors
Author Response
Dear Reviewer,
Thanks very much for taking your time to review our manuscript (Manuscript ID: insects-1830806). We really appreciate all your comments and suggestions! We have carefully considered your suggestion and tried our best to revise our manuscript according to the comments.
All changes have been highlighted in new version of manuscript and the following are the responses.
Response to the comments of Reviewer #3
Comment No.1: This draft needs a rigorous grammar and English-proof revision before its publication. This version does not read easy, and have several writing errors and typos that need to be fixed.
Response: The manuscript has been checked by a native speaker.
Comment No.2: I recommend to include and briefly comment on the potential evolutionary consequences, as it is being found both in arthropods and vertebrates (e.g. humans), that the pollution by these heavy metals may trigger epigenetic molecular mechanisms involved in long stress responses. These can have transgenerational consequences, either promoting the adaptation of these organism to the contaminated environment, or participating in teratogenic or pathological responses. In this context I recommend you that when introducing the toxic effects of heavy metals on insects and invertebrates please cite:
Olivares-Castro, G., Cáceres-Jensen, L., Guerrero-Bosagna, C., & Villagra, C. (2021). Insect epigenetic mechanisms facing anthropogenic-derived contamination, an overview. Insects, 12(9), 1–29. https://doi.org/10.3390/insects12090780
And regarding Cd toxic effects on humans:
B. Wang, Y. Li, C. Shao, Y. T. and L. C. (2012). Cadmium and its epigenetic effects. Current Medicinal Chemistry, 19(16), 2611–
Response: Many thanks for your valuable suggestions and literatures recommended. We have improved and revised the manuscript according to your suggestions mentioned above. (New version of manuscript, in line 58-60, line 275-278, and line 284-287)
Thanks again to the hard work of the reviewer!
